# Grounding Terms from an Ontology for use in Autoformalization: Tokenization is All You Need

**Richard Thompson**                             RICHARD.THOMPSON@NPS.EDU
**Angelos Toutsios**                          ANGELOS.TOUTSIOS.GR@NPS.EDU
**Adam Pease**                                     ADAM.PEASE@NPS.EDU
**Mathias Kolsch**                                    KOLSCH@NPS.EDU
*CS Dept. Glasgow Hall East, Building 305 1411 Cunningham Rd Monterey, CA, 93943*

**Editors:** Leilani H. Gilpin, Eleonora Giunchiglia, Pascal Hitzler, and Emile van Krieken

## Abstract

Large Language Models (LLMs) have shown strong performance in translating natural language into programming languages like Python or Java. However, for niche computer languages, where there is limited training data, fine-tuning a base model is often necessary. A key challenge arises when the pretrained embeddings of natural language terms interfere with the intended syntax and semantics of formal language terms. This issue is especially pronounced in the logical language of SUO-KIF, which is used in the Suggested Upper Merged Ontology (SUMO). SUMO contains thousands of terms that closely resemble everyday English words. As a result, models often produce syntactic errors or hallucinate non-existent terms due to conflicting embeddings learned during base training.

This work introduces a tokenization-based technique to mitigate these issues. By altering how formal terms are tokenized, we can decouple their embeddings from similar natural language words, significantly reducing syntax errors and term hallucinations in the generated formal language output.

## 1. Introduction

Formalizing natural language into logic-based representations is a long-standing goal in artificial intelligence, as a way to enable machines to reason about the world. While recent large language models (LLMs) have excelled at translating natural language prompts into programming languages like Python or Java, less attention has been paid to niche formal languages that support knowledge representation and logical inference.

One such language is SUO-KIF, a higher-order logic language used in the Suggested Upper Merged Ontology (SUMO).[1] The language and ontology together allow for the formal specification of entities and their attributes, enabling automated reasoning. For example, the sentence *"The apple is red"* may be formalized as:

```
(exists (?C) (and (instance ?C Apple) (attribute ?C Red)))
```

Although the logical language is quite simple, with keywords for just a few logical operators (and, or, not, forall, exists, $\Leftrightarrow$, $\Rightarrow$ and equals), we also must ensure that logical statements use the correct terms from the SUMO library, and there are approximately 20,000 of these. The problem may be considered similar to not only generating the correct

---

1. SUO-KIF specification, SUMO, and source code is found at https://github.com/ontologyportal

syntax for Python or Java, but also selecting the right function or method names from tens of thousands of possible choices in libraries.

Another challenge that arises is that SUMO terms often resemble everyday English words. Despite fine-tuning, LLMs frequently produce syntax errors and hallucinated terms due to statistical associations learned during pretraining. LLMs break input into subword tokens and embed them in a semantic space, where related terms cluster, as shown simplistically in Figure 1.

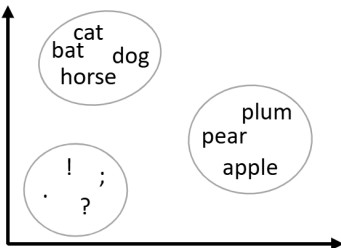

Figure 1: Simplistic view of an embedding space, where similar tokens are grouped together.

Strong embeddings can create a problem when translating to SUO-KIF. For example, variables begin with "?" and have no whitespace in SUO-KIF, while English treats "?" as punctuation. Sentences translated from English often have a space incorrectly inserted after the "?" in SUO-KIF. Similarly, models may confuse formal SUMO terms like `BatMammal` with invalid hallucinated ones like `BatAnimal`, driven by natural language co-occurrence.

To address this, we introduce a method to re-ground formal SUMO terms in a way that decouples them from natural language embeddings. SUMO terms are mapped to new, unrelated tokens. During the fine-tuning process these new terms are grouped appropriately in the embedding space, but lack the undesired positional encodings and statistical correlations of previous tokens. This reduces both syntax errors and term hallucinations, significantly improving SUO-KIF generation quality.

## 2. Related work

Researchers have shown fine-tuning to be an effective way to improve generation of programming language statements (Shypula et al., 2024). Progress has also been made in auto-formalization of mathematical problems that have been expressed semi-formally (Wu et al., 2022). In translating from one language to another, improper tokenization has been shown to increase the rate of hallucinations of terms (Wang et al., 2024).

Ontology alignment and grounding have traditionally focused on matching human concepts across knowledge bases, but not on decoupling formal representations from their natural language counterparts within generative models. Previous work in semantic grounding, such as retrofitting embeddings using ontology structures (Faruqui et al., 2015) or concept embeddings from WordNet (Camacho-Collados et al., 2016), aligns knowledge bases based on semantic similarity. However, these methods still preserve natural language proximity and do not address the situations where that proximity is problematic.

Several approaches have been proposed to handle domain-specific vocabularies in neural language models. Vocabulary expansion techniques add domain-specific terms directly to the model's token vocabulary (Toraman et al., 2023), but this approach significantly increases memory requirements, computational cost, and training time. Additionally, when domain terms closely resemble existing vocabulary (as is often the case with SUMO terms), vocabulary expansion only mitigates the embedding interference problem.

## 3. Background

### 3.1. SUO-KIF

SUO-KIF (Standard Upper Ontology—Knowledge Interchange Format) is a formal language designed for expressing ontological and logical statements in a machine-readable way (Pease, 2009). Unlike more widely adopted languages such as the OWL family of description logics, SUO-KIF provides a more expressive higher order logic (Benzmüller and Pease, 2010). It is particularly suited for encoding complex conceptual knowledge.

### 3.2. SUMO

The Suggested Upper Merged Ontology (SUMO) is a comprehensive formal ontology originally intended to provide a foundation for more specific domain ontologies (Niles and Pease, 2001; Pease, 2011). Developed to support automated reasoning, SUMO includes thousands of terms and axioms written in SUO-KIF that describe abstract concepts (e.g., `Attribute`, `Quantity`) and concrete entities (e.g., `Apple`, `Book`).

### 3.3. T5 and Flan-T5

T5 (Text-to-Text Transfer Transformer) and its variant Flan-T5 are transformer-based language models that frame NLP tasks as text generation problems (Raffel et al., 2023). In our work, we experiment with both T5 and Flan-T5 as the base models for translating informal English prompts into formal SUO-KIF expressions. Trained on a wide range of tasks using a unified format, T5 and Flan-T5 have shown strong performance across translation benchmarks, and are considered state of the art (Longpre et al., 2023). Tokenization was conducted using the native T5Tokenizer, which is based on the popular SentencePiece tokenizer (Kudo and Richardson, 2018).

**A Note on vocabulary expansion:** An alternative solution to explore is vocabulary expansion, where whole SUMO terms are explicitly added to the token set, and then selection of only these tokens is forced in the output. This approach has been shown to significantly increase memory requirements, cost, and training time (Toraman et al., 2023). Additionally, many SUMO terms align exactly with existing tokens, thus vocabulary expansion mitigates, but does not eliminate the issue.

## 4. Methodology

Training data consisted of approximately 6 million English sentences with their SUO-KIF logic equivalent. Both SUO-KIF key words, variables, and unique SUMO terms were extracted from the training data and assigned a corresponding label composed of five random

capital letters. An example is shown in Table 1. For example, the SUMO term `Historian` was mapped to `AJOFN`. In this way, we semantically separate the SUMO term from its English token, without a strong grounding in the model. During training, the new mappings are moved to appropriate locations in the embedding space. Characters such as parentheses and whitespace were not translated, as the base model already handles them properly.

Table 1: An example of re-grounding

| English sentence | The historian is not awake right now. |
|---|---|
| Logic translation | (not
  (exists (?H)
    (and
      (attribute ?H Historian)
      (equal ?T Now)
      (holdsDuring ?T (attribute ?H Awake))))) |
| Re-grounded translation | (SIRQJ
  (LOAXA (UGQJQ)
    (QAGRM
      (RJGUO UGQJQ AJOFN)
      (ANNFF KABBQ OFHBH)
      (ILEGC KABBQ (RJGUO UGQJQ LZEJO))))) |

Fine-tuning is conducted using both the Flan-T5 and T5 model. A baseline model was fine-tuned for each, using the English sentences and the normal logic translations for training data. Re-grounded models were also trained for both Flan-T5 and the T5 models, using the English sentences and the re-grounded translations.

After training the models, testing was conducted on a standard set of 100 test sentences, chosen for their grammatical diversity, that remained constant through the experiment. Additionally, testing was conducted with 100 sentences randomly chosen from a corpus of policy documents comprising approximately 60,000 sentences. For the models trained using re-grounded SUMO terms, postprocessing was conducted to translate the model output back to the corresponding SUMO terms.

## 5. Results

Results using a static set of sentences are shown in Table 2. Results with sentences pulled randomly from a large corpus are shown in Table 3. A translation was considered syntactically correct if it followed the SUO-KIF grammar rules, without regard to type accuracy. A translation was considered type correct if all terms had a corresponding term in SUMO.

An important caveat to the baseline data is that the syntactic accuracy was only achieved with postprocessing by ensuring there was white space before the "?" character, and removing any white space between it and the variable name. Without this postprocessing

Table 2: Percent of sentences translated correctly on standard test set.

| (a) Baseline | | | | (b) SUMO Terms Re-grounded | | |
|---|---|---|---|---|---|---|
| | **T5** | **Flan-T5** | | | **T5** | **Flan-T5** |
| Syntax correct | 45% | 72% | | Syntax correct | 94% | 93% |
| Types correct | 8% | 9% | | Types correct | 66% | 74% |
| Both correct | 7% | 8% | | Both correct | 61% | 70% |

Table 3: Percent of sentences translated correctly on randomized test set.

| (a) Baseline | | | | (b) SUMO Terms Re-grounded | | |
|---|---|---|---|---|---|---|
| | **T5** | **Flan-T5** | | | **T5** | **Flan-T5** |
| Syntax correct | 53% | 58% | | Syntax correct | 90% | 85% |
| Types correct | 3% | 4% | | Types correct | 63% | 74% |
| Both correct | 3% | 4% | | Both correct | 59% | 62% |

step, not a single sentence would be syntactically correct when using either baseline model. This postprocessing step is not conducted (or needed) for the re-grounded models.

Re-grounding SUMO terms had a significant positive impact in both the T5 and Flan-T5 models. Type accuracy improved greatly, and fewer terms were hallucinated. It is hypothesized that greater grammar and vocabulary diversity in the training data will further reduce remaining type errors. Future work should include a study on how the extent of term re-grounding affects the semantic accuracy of the translation. While it did not appear to have had much impact, the impact will become clearer as the diversity of the training set grows.

## 6. Conclusion

This study demonstrates that formal terms that overlap semantically or syntactically with everyday English significantly hinder the accurate translation of natural language into formal languages like SUO-KIF. We introduced a simple yet effective tokenization-based re-grounding technique that disconnects formal SUMO terms from their natural language embeddings. By mapping these terms to randomized token sequences, we reduce the influence of pretrained embeddings and improve both syntactic and type accuracy. Our experiments with T5 and Flan-T5 models show that this approach dramatically outperforms baseline fine-tuning in terms of correctness and robustness. This method itself is model agnostic, computationally lightweight, and requires no vocabulary expansion. Future work will explore its effects on semantic fidelity and extend its application to more diverse language inputs.

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
