# OpenReview forum: "Grounding Terms from an Ontology for use in Autoformalization: Tokenization is All You Need"
_nesyconf.org/NeSy/2025/Conference_Phase_2 — NeSy 2025 - Phase 2 Poster_

### Official Review · Reviewer_gAGT · 2025-06-24
**Improving auto-formalization for formal languages by re-grounding**

**Rating:** 6
**Confidence:** 4

**Review:**

This paper aims for the automated translation of natural language sentences into a formal language (SUO-KIF) using large language models (LLMs). The SUO-KIF language appears to be particularly difficult to translate with present LLMs, since many expressions needed in SUO-KIF collapse with common tokens used by the tokenizer. To this end, the paper proposes to re-ground all formal terms into new, unrelated tokens. Fine-tuning a T5 (or Flan-T5) on these re-grounded translations yields significantly better results, reducing syntactic errors as well as hallucinations.

This paper tackles an important problem in auto-formalization methods. LLMs (and particularly their tokenizers) cannot be trained/designed for all niche formal languages. This paper proposes an approach to circumvent potential collisions in the token space that result in LLM’s hallucinations.

As the main downside, the mapping requires full retraining on all possible terms. Hence, the generalization of new, unseen terms is not possible; it requires retraining. Hence, the method cannot really be attributed to being “computationally lightweight”. For improving the generality of the approach, I suggest only mapping certain terms to unrelated tokens (not, exists, etc.) while others remain (Historian, Awake, etc.). This would allow for lexical generalization.

**Anonymity:**

Remain anonymous

---

### Official Review · Reviewer_dTTx · 2025-07-03
**Good paper but lacking a stronger relationship to NeuroAI**

**Rating:** 6
**Confidence:** 4

**Review:**

This paper tackles the challenge of accurately auto-formalizing English sentences into SUO-KIF by re-grounding SUMO ontology terms via tokenization. The authors map each SUMO term to a randomized token sequence, then fine-tune T5 and Flan-T5 on paired English/SUO-KIF data. Evaluations on a held-out 100-sentence test set and 100 sentences from policy documents show significant gains.

Major points/comments.

The experiments show a significant improvement, re-grounded models nearly double both syntactic and ontological correctness across two distinct test sets.

The approach is based on a simple idea and it is efficient, indeed token remapping is computationally light and conceptually straightforward.

The approach is model-agnostic, and applicable to any subword-based LLM, suggesting broad utility for niche formal languages beyond SUO-KIF.

The paper is well-written and, although is a short paper, it is self-contained.

However, the paper misses a stronger relationship to neuroAI. It presents a purely neural intervention without articulating how the symbolic ontology structure is leveraged beyond token decoupling. There is no discussion of how this contributes to the neurosymbolic integration of symbolic knowledge (SUMO) with neural inference (LLM).

**Anonymity:**

Remain anonymous

---

### Official Review · Reviewer_qrCS · 2025-07-05
**The authors propose an approach based on LLMs for translating natural language into SUO-KIF.**

**Rating:** 4
**Confidence:** 5

**Review:**

1- Positive points
Niche computer languages such as SUO-KIF lack datasets for training ML models. In addition, many LLMs are not specialized in the translation of natural language into these languages.

2-Negative points
- The fine-tuning method is not the only way used today to obtain results from LLMs. The authors did consider (or even present) the RAG-based approach.
- The methodology section is not reproducible (see the questions).
- Many claims in the paper without examples to show how any LLM of the choice of the authors hallucinate
- The authors did not describe the evaluation well. The evaluation of the generated output using BLEU and ROUGE score was expected. The evaluation metrics used by the authors are not well described

3- Questions
- Where does the training set come from?
- The data was split into train tests?
- Which fine-tuning method was used?
- How was the test set obtained?

4- Recommendations
Put the related work after the background
- “While recent large language models (LLMs) … or Java”: references should be provided to support this claim
- Provide citations during the first use of SUMO and SUO-KIF in the introduction
- “Despite fine-tuning, … during pretraining”: Provide examples or references to support this claim
- “Strong embeddings … SUO-KIF”: Provide examples of hallucinations with SUO-KIF with existing LLMs of your choice

**Anonymity:**

Remain anonymous